# Local Endoscopic Treatment of Low-Grade Urothelial Tumors in the Upper Urinary Tract—Oncological Outcome of a Consecutive Single-Center Series of 118 Patients

**DOI:** 10.3390/cancers16223795

**Published:** 2024-11-11

**Authors:** Sophie Bislev, Simone Buchardt Brandt, Sven Fuglsig, Jørgen Bjerggaard Jensen

**Affiliations:** 1Department of Clinical Medicine, Aarhus University, 8000 Aarhus C, Denmark; simbra@rm.dk (S.B.B.); bjerggaard@skejby.rm.dk (J.B.J.); 2Department of Urology, Aarhus University Hospital, 8200 Aarhus N, Denmark; svenfugl@rm.dk

**Keywords:** laser, ureteroscopy, uteroscopy instrumentation, laparoscopy benign disease, laparoscopy malignant disease

## Abstract

This paper presents the results of our retrospective cohort single-center study. Our findings demonstrate that patients diagnosed with low-grade upper tract urothelial tumors exhibit a high recurrence rate when treated with laser ablation, but a low progression rate. While the overall mortality in this cohort is relatively high, deaths related specifically to upper tract urothelial tumors are low. Importantly, our data indicate that recurrences primarily occur within the first few years after diagnosis. Thus, vigilant follow-up during this period proves effective in detecting recurrences. We conclude that laser ablation stands as a safe and less invasive alternative to radical nephroureterectomy.

## 1. Introduction

Upper tract urothelial cancer (UTUC) is a rare type of urothelial cancer with 200 cases a year nationally, and it only represents 15% of all urothelial cancers with the remaining to be found in the urine bladder. The tumor derives from the urothelial and 2/3 are localized in the pelvis and 1/3 in the ureter [1,2].

The European Association of Urology (EAU) guidelines recommend local treatment of upper tract urothelial cancer (UTUC) with laser ablation instead of radical nephroureterectomy (RNU) for patients with low-grade (LG) unifocal tumors or in patients where RNU is contraindicated and local treatment is possible, as RNU is known to be a major operative procedure that includes a number of post-operative and long-term complications [3,4].

Previous studies find no difference in cancer-specific survival (CSS) when comparing local treatment with RNU. These studies include all stages of UTUC. The risk of cancer-specific death was higher in patients with high-grade (HG) UTUC treated locally, and the overall survival (OS) was better in patients with LG UTUC treated locally [5,6,7,8,9,10].

The kidney-sparing treatment includes the risk of local recurrence. Previous studies have found between 68% and 92% of the patients develop a local recurrence, with an average of more than three recurrences pr. patient [7,11,12].

Previous studies either compare kidney-sparing surgery with radical nephroureterectomy in treatment of both LG and HG tumors regarding survival and recurrence. More specific studies with more narrow inclusion criteria regarding tumor stage and tumor grade are needed.

In the present study, we aimed to investigate the long-term outcomes for patients treated with endoscopic kidney-sparing surgery in patients with pTa og pT1 LG UTUC.

## 2. Methods

### 2.1. Patient Recruitment

All patients with pTa low grade (LG) and pT1 LG tumor in the upper urinary tract in the period from 2012–2022 in the Central Region Denmark, which has a population of approximately 1.3 million habitants, were included. Patients were identified through the pathology code registered in the regional pathology database by SNOMED codes registered at the time of pathology examination [13]. The patients were treated in two different public regional hospitals each with their own pathology department.

Individual patient data were obtained retrospectively from electronic patient medical charts. The creatinine clearance was calculated from creatinine level, gender, height, and weight using the Cockcroft–Gault equation [14]. Comorbidities were registered as the Charlson Comorbidity Index [15]. The size of the initial tumor was found either in the CT-scan description or in the surgical note.

All data were collected and managed using REDCap electronic data capture tools version 14.5.16 hosted at the Institute of Clinical Medicine, Health, Aarhus University [16].

### 2.2. Diagnosis and Treatment Plan

All patients with suspected tumor of the upper urinary tract underwent ureterorenoscopy (URS) with biopsy, unless nephroureterectomy was indicated already or metastases were identified at the time of diagnosis. If the tumor was papillary with no suspicion of invasion, it was most often treated during the same procedure with laser ablation. The laser used was an Olympus Soltive Thulium laser (Tokyo, Japan) with a 150 nm fiber. After the first local treatment, patients with Ta LG tumor underwent a 4-month follow-up control with URS. In cases of no local recurrence, new URS was performed after additional 4 and 8 months and hereafter yearly control CT-urography for three years. Cystoscopy to detect bladder recurrence was performed at all follow-up visits. In patients where local treatment was chosen due to other indications than Ta LG, the follow-up was based on an individual evaluation [2,5,17].

### 2.3. Statistical Analysis and Outcomes

The Nelson–Aalen plot was used to illustrate the cumulative hazard of getting first local recurrence. A Kaplan–Meier curve was constructed with the same data to illustrate the time in which the local recurrence occurred after diagnosis.

Progression was defined as a progression from pTa LG and included pTa HG. The event was either death, pathological progression to higher stage, or RNU.

Progression-free survival, OS, and CSS were analyzed using Kaplan–Meier curves with standard deviation of the median and interquartile range (IQR). Follow-up time was defined as time from histology verified diagnosis to death or the date of last data entry. Death was noted as either UTUC specific or other cause of death. 

The statistical analysis was performed in Microsoft Excel version 16.72 using XLSTAT free trial version. 

### 2.4. Ethical Approvement

The study was approved by the Central Denmark Region Committees on Health Research Ethics (approval ID: 1-45-70-36-23). Signed consent was waived from our ethical committee in patients not alive at the time of data collection, whereas written informed consent was obtained from all living participants involved.

## 3. Results

### 3.1. Patient Characteristics

A total of 324 patients fulfilled the pathological SNOMED code criteria for LG NMIBC. Of these, 206 patients were excluded from further analysis for various reasons (Figure 1). This resulted in 118 patients with LG UTUC who underwent initial endoscopic local treatment (Table 1).

### 3.2. Local Recurrence

The median time from diagnosis to first local recurrence was 5 months (IQR: 2–46). The cumulative HR of getting local recurrence is illustrated in Figure 2. The total follow-up time was 122 months (10.2 years) and the cumulative hazard at this point was 2.0 (IQR: 0.9–3.0) (SE = 0.54). At 12 months (1 year), the HR was 1.0 (SE = 0.13).

One year after initial treatment, 33.8% (SE = 0.045) were local recurrence free (Figure 3). The 2-year RFS was 28.8% (SE = 0.045).

Recurrence characteristics are shown in Table 2.

### 3.3. Progression

In the present study, 20 out of the 118 patients had local recurrence with progression, and of those, 8 proceeded to RNU. In total, 20 of the 118 patients had salvage RNU (Table 3). The median survival time with no progression or RNU was 48 months (2 years). The 2-year progression and RNU-free survival distribution was 67.9% (SE = 0.044) and the 5-year progression and RNU-free survival was 44.5% (SE = 0.054). The survival distribution is illustrated in Figure 4. The median follow-up time was 26 months (IQR: 11–53).

### 3.4. Long Term Survival

The median follow-up time was 36.0 months (IQR: 20.3–58.8). Figure 5A illustrates the OS and the CSS. The OS after a total follow-up of 141 months (11.8 years) was 38.9% (SE = 0.074). The 2-year OS was 84.2% (SE = 0.035) and the 5-year OS was 59.1% (SE = 0.057). The median OS time was 88 months (7.3 years) after diagnosis.

The DSS was 94.1% (SE = 0.026) at end of follow-up (141 months); no patients had a disease-specific death after 36 months of follow-up, illustrated in Figure 5B. The 2-year DSS was 97.1% (SE = 0.017). The median disease-specific mortality time was 20 months (IQR: 12–27).

## 4. Discussion

In the present patient series, we found that the local recurrence rate following local laser ablation in LG UTUC was high. However, we also found that risk of progression was low, and that the CSS was clearly higher than the OS. Thus, we found that the mortality is predominantly related to other causes than the urothelial tumor following local laser ablation, indicating that it is a safe procedure in selected patients.

Seisen et al. found no difference in CSS when comparing local treatment with RNU in a systematic review from 2016 [5]. They defined local treatment as both segmental ureterectomy and endoscopic procedures. In five of the included twenty-two studies from the meta-analysis, endoscopic treatment was compared with RNU. Here, no significant difference in 5-year and 10-year CSS was found between endoscopic treatment and RNU. The studies included all T-stages of UTUC. They found that the risk of cancer-specific death was higher in patients with HG UTUC who received endoscopic treatment, whereas the OS was significantly better in the patients with LG UTUC undergoing endoscopic treatment. These findings are contradicted in a retrospective study by Kim et al., who found that OS and CSS after 24 months were similar when treated endoscopically compared with RNU, but after 24 months, the OS and CSS were lower for patients treated endoscopically. However, the group of endoscopically treated patients had a large heterogeneity since comorbidities were more frequent and the patients were older, for which the investigators did not adjust for [6].

The OS found in this study is the OS of patients with a diagnosis of LG UTUC treated with laser ablation initially. The OS is relatively low and could be due to the high mean age at diagnosis. Furthermore, this study finds a higher recurrence rate than other similar studies. This could be due to more patients being selected for local treatment over RNU in this study than in other studies [7,11,12]. Another bias that could have underestimated the OS in the present study is that potential heredity was not obtained from the patients. A study shows that 5.2% of the patients having UTUC had confirmed Lynch syndrome, and these patients have higher risk of specific cancer types [18].

The local recurrence and median time from diagnosis to first local recurrence are lower in the present study compared to those reported in Shenhar et al., where 92% had local recurrence with an average of 3.2 recurrence pr. patient. This could be due to the difference in inclusion criteria, where the patients in the present study had lower stages of tumors. In the present study, 24.1% underwent salvage RNU after initial local treatment, compared to only 17% reported by Shenhar et al., despite the patients in the present study having lower stages. However, this discrepancy could reflect the differences in national guidelines of when to perform RNU initially [11].

The risk of local recurrence following endoscopic treatment of UTUC is important to consider when offering endoscopic treatment. We found that the risk of local recurrence was highest within the first 18 months following initial treatment. The currently recommended control program after diagnosis recommends follow-up for a minimum of 5 years but often lifelong [3]. After 5 years, the cumulative incidence increases from 1.5 to 2.0 in our present series; however, this is based on only one patient who had a first local recurrence after 5 years. This could indicate that the current control program is sufficient for diagnosing local recurrence and could be terminated following a 5-year recurrence-free interval and only be undergoing new investigations in case of symptoms [3]. The current national guidelines of treatment of urothelial tumors in the upper urinary tract are based on retrospective studies, where local treatment is compared with RNU or studies where the outcomes of local treatment are illustrated. In these studies, no significant difference was found in oncological outcomes [19].

The EAU recommends RNU in cases of HG UTUC [17]. Studies have found the disease-free OS to be between 66% and 71% and the 5-year DSS to be between 72% and 75% [20,21]. The studies only include HG UTUC. Compared to this study, especially the DSS is much lower than the DSS found in the present study. The outcomes of these studies are difficult to compare directly, as the patient groups included are not comparable due the difference in disease severity. If compared, endoscopic treatment is related to a better CSS than RNU, but a large bias in comparison is the patient groups, as HG UTUC is known to have a lower CSS than LG UTUC [22,23]. Other kidney-sparing treatment methods for UTUC located in the ureter are segmental ureteral resection or percutaneous access. This is recommended for LG UTUC that cannot be removed endoscopically [17,24]. Studies show no significant difference in short-term outcomes comparing RNU and segmental ureteral resection, but these methods are used less due to improved endoscopic tools [24,25,26]. A study investigating the oncological outcome after percutaneous endoscopic resection showed a 5-year DSS of 79.5%, which is lower than that found in the present series, indicating that the endoscopic procedure is associated with higher survival. The bias of this study was the low number of included patients (n = 40) [27].

The present study is limited by the retrospective design and the absence of continuous monitoring throughout the follow-up period from the patient’s last visit to the department until data entry; however, the risk of local recurrence hereafter is low. Furthermore, the data were collected from the regional chart system; however, not all data were accessible and were simply not registered. An important factor that lacked in a series of patients in the present study was the size of the initial tumor. The present study stands out from other studies in the current literature due to the large number of patients included [2,5,7].

Retrospectively, a series of data was not obtained from the patients, which could have been interesting to include in the analysis. An important factor that was missed was which of the patients had previous bladder cancer, as bladder cancer is known as a risk factor of getting UTUC [28,29]. This could underestimate the OS, as 30 patients from this study had a history of bladder cancer. A subgroup analysis should have been performed to exclude the 40 patients who had a history of bladder cancer. Another factor that we missed was the marital status of the patients, as this could have had an impact on the effect of the treatment pattern.

Although the number of patients in this study is relatively large, if a larger cohort size is desired, the same kind of study could be conducted in 5–10 years. The next step in this research could be to conduct a study where the two treatment methods, local treatment of RNU, for these types of patients could be compared to see if the patients who undergo RNU have better outcomes than patients treated locally. Another outcome parameter in this kind of study could be the development of kidney function after treatment alongside survival distributions. Furthermore, the effect of treatment with BCG or mitomycin C was not investigated in this study. In further research, it could be interesting to determine if it has the same effect as in treatment of bladder tumors [30].

## 5. Conclusions

In conclusion, patients diagnosed with LG UTUC tumors have a high local recurrence rate but a low rate of progression. We find that close post-operative follow-up for the first few years is indicated to detect early local recurrences. The overall mortality rate is high in this cohort; however, only a small fraction of deaths is related to UTUC after initial local treatment. We find that local ablation is a safe and less invasive alternative to up-front RNU, which is associated with high DSS for treatment of UTUC.

## Figures and Tables

**Figure 1 cancers-16-03795-f001:**
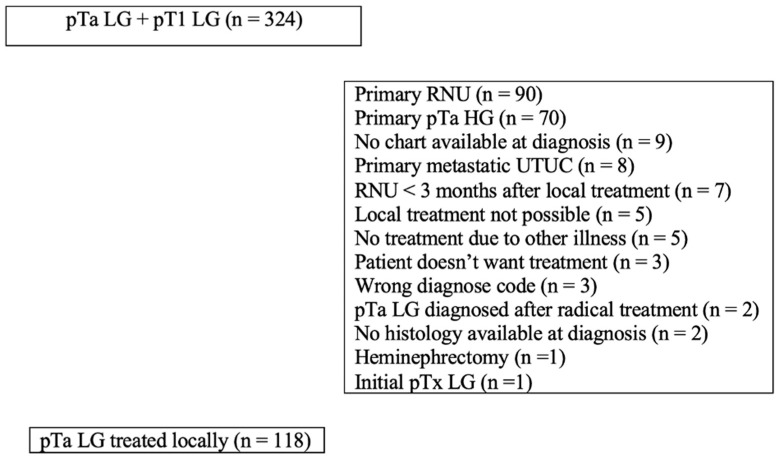
Flow chart showing the inclusion and exclusion of patients.

**Figure 2 cancers-16-03795-f002:**
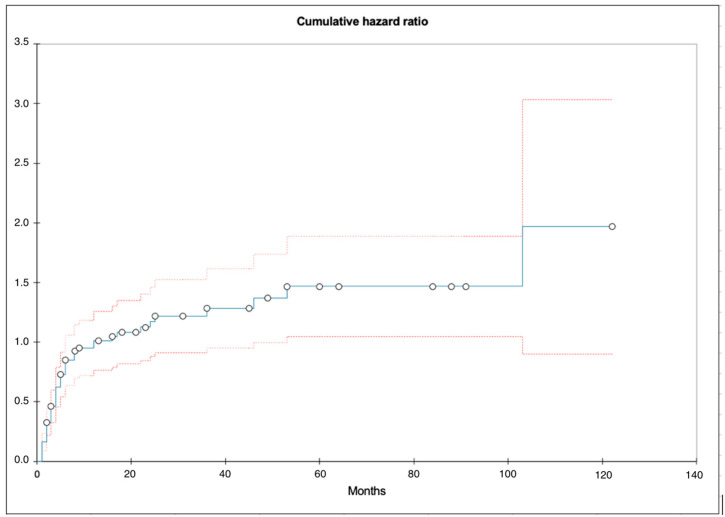
Nilsson–Aalen plot illustrating the hazard ratio of getting a local recurrence after diagnosis. Blue curve = median hazard, upper red curve = 75% quantile, lower red curve = 25% quantile, white spots = event or censored.

**Figure 3 cancers-16-03795-f003:**
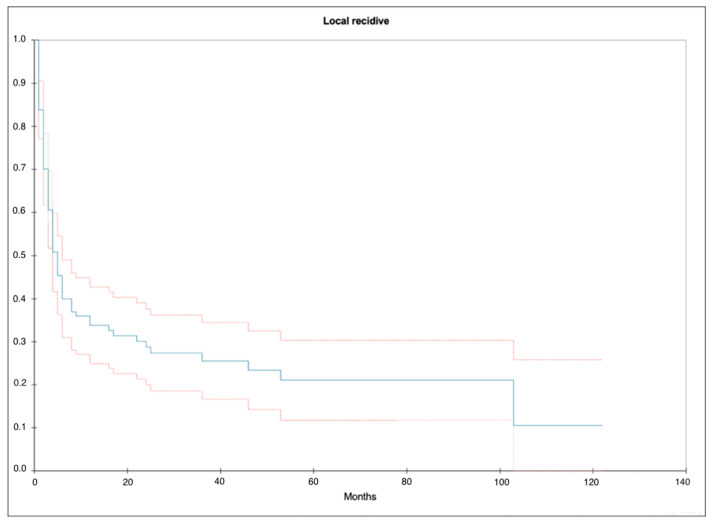
Kaplan–Meier curve illustrating the local recurrence distribution from diagnosis to first local recurrence. Blue curve = median, upper red curve = 75% quantile, lower red curve = 25% quantile.

**Figure 4 cancers-16-03795-f004:**
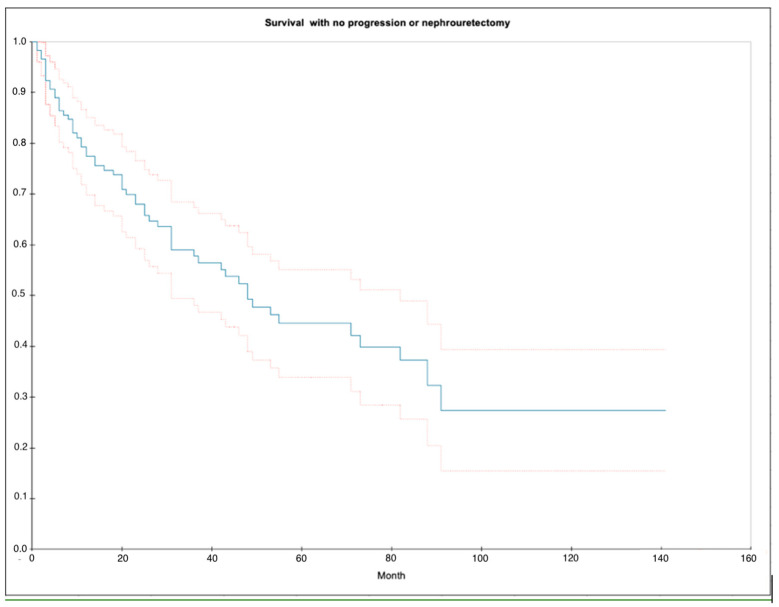
Kaplan–Meier curve illustrating the survival distribution of surviving with no progression and no nephroureterectomy. Blue curve = median, upper red curve = 75% quantile, lower red curve = 25% quantile.

**Figure 5 cancers-16-03795-f005:**
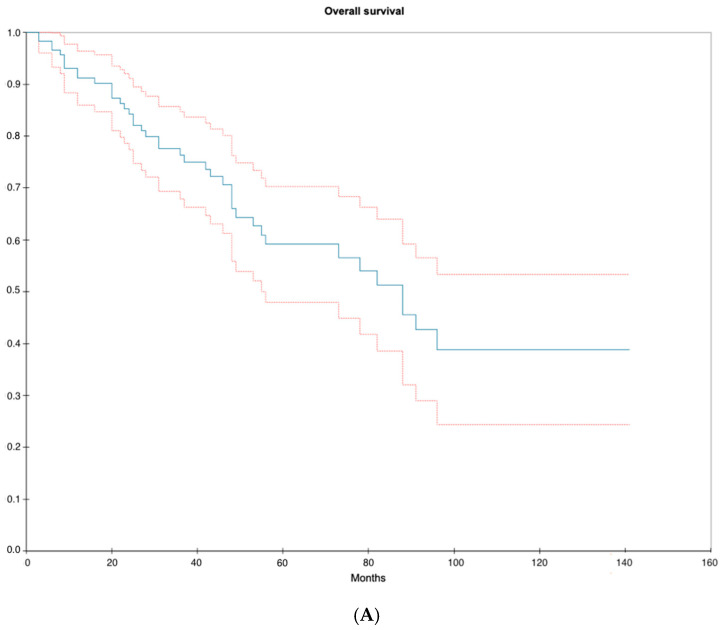
(**A**): Kaplan–Meier curve illustrating overall survival from diagnosis to death or date of data collection. Blue curve = median, upper red curve = 75% quantile, lower red curve = 25% quantile. (**B**): Kaplan–Meier curve illustrating disease-specific survival from diagnosis to death from disease or data collection. Blue curve = median, upper red curve = 75% quantile, lower red curve = 25% quantile.

**Table 1 cancers-16-03795-t001:** Patient demographic and characteristics and information regarding status at diagnosis and tumor information. N = number, ASA = severe systematic disease, LG = low grade, HG = high grade, NA = not available, RNU = radical nephroureterectomy.

Sex (male/female), n (%)	82 (69.5)/36 (30.5)
Age (years), median (IQR)	75 (69–81)
Body mass index (kg/m^2^), median (IQR)	26 (23–28)
No BMI available, n	24
ASA grade, n (%)	
1	8 (6.8)
2	61 (51.7)
3	46 (39.0)
4	2 (1.7)
5	0 (0.0)
NA	1 (0.8)
Charlsons Comorbidity Index, n (%)	
1	26 (26.5)
2	28 (28.6)
3	22 (21.4)
4–5	12 (12.2)
6–7	8 (8.2)
8–9	3 (3.1)
Smoking, n (%)	
Never	42 (35.6)
Former	28 (24.6)
Present	44 (37.3)
NA	3 (2.5)
History of bladder tumors (yes/no), n (%)	40 (35.6)/78 (64.4)
Treatment with cystectomy (yes/no), n (%)	5 (12.5)/35 (87.5)
Former contralateral RNU (yes/no), n (%)	8 (6.8)/110 (93.2)
Diagnosis characteristics	
Hydronephrosis, n (%)	
Yes	23 (19.5)
No	92 (78.0)
NA	3 (25)
Creatinine (μmol/L), median (IQR)	86.5 (73–108)
Creatinine clearance (μmol/L), median (IQR)	63.0 (43.9–79.8)
Number of tumors, n (%)	
1	95 (80.5)
2	18 (15.3)
>2	3 (2.5)
NA	2 (1.7)
Location of tumor, n (%)	
Right	60 (50.8)
Left	54 (45.8)
Bilateral	4 (3.4)
Tumor stage, n (%)	
pTa LG	116 (98.3)
pT1 LG	0 (0.0)
pTis LG	2 (1.7)
Size of largest tumor (mm), median (IQR)	15 (10–20)
Urine cytology at diagnosis n (%)	
LG	78 (66.1)
HG	4 (3.4)
NA	36 (30.5)
JJ-catheter following procedure (yes/no), n (%)	113 (95.8)/5 (4.2)
Acute re-hospitalized within 90 days after procedure (yes/no), n (%)	21 (17.8)/97 (82.2)

**Table 2 cancers-16-03795-t002:** Recurrence information. n = number, BCG = bacille Calmette-Guerin.

Recurrence	
Follow-up duration (months), median (IQR)	36.0 (20.3–58.8)
Local recurrence in upper urinary tract (yes/no), n (%)	83 (69.3)/35 (29.7)
Local recurrence with progression, n (%)	20 (24.1)
Treatment with mitomycin C, n (%)	8 (9.6)
Treatment with BCG, n (%)	5 (6.0)
Recurrence in the contralateral urinary tract, n (%)	10 (12.0)
Recurrence, n (%)	
In bladder	52 (44.1)
Metastatic	5 (4.2)
Number of recurrence pr. Patient, median (IQR)	2 (1–5)
Time to first recurrence (months), median (IQR)	5 (2–46)

**Table 3 cancers-16-03795-t003:** Outcome information. n = number.

Initial pTa LG, n (%)	118 (100)
Salvage nephroureterectomy, n (%)	20 (16.9)
Death, n (%)	42 (35.6)
Death from other cause, n (%)	37 (31.4)
UTUC-specific death (yes/no), n (%)	5 (4.2)

## Data Availability

The datasets presented in this article are not readily available because of Danish law (general data protection regulation (GDPR)) regarding personal information that says this information cannot be handed out without legal permission.

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
