# Peer review of "Local Endoscopic Treatment of Low-Grade Urothelial Tumors in the Upper Urinary Tract—Oncological Outcome of a Consecutive Single-Center Series of 118 Patients"

_cancers, 2024, doi:10.3390/cancers16223795_

Round 1

Reviewer 1 Report

Comments and Suggestions for Authors

The authors should be congratulated for their work. The present study aimed to investigate the oncological outcome of 11 patients treated endoscopically for low-grade (LG) upper tract urothelial carcinoma (UTUC). The topic is novel and there is an urge to collect new pieces of knowledge on that topic, due to the rarity of the UTUC as well as the low adherence to focal therapy in this cancer. 

The study relied on 118 UTUC patients from a single center institution. 

Briefly, they found that the median time to first recurrence was 5 months Moreover, the two-year overall survival (OS) and disease-specific survival (DSS) were 84.2% and 97.1%. The 5-year OS and DSS were 59.1% and 94.1%. Those results according to the authors suggested the feasibility of controlling DSS even if a local therapy strategy.

Unfortunately, such conclusions are speculative. The mortality in these patients is invariably biased by their older age. More weight is attributable to other-cause mortality rather than UTUC-specific deaths. Moreover, history of bladder cancer also may affect the results. A subgroup analysis should be performed to remove those patients. Additionally, nephroureterectomy in the table 3 is confusing. Was it delivered after the local treatment or upfront in this low-grade patients? If it was performed upfront a comparison between those groups should be added. Moreover, a competing regression risk model should be performed to establish the role of local therapy vs. RNU in the DSS and OS of these UTUC subjects. Any data on the marital status of these patients? It may have an effect on their treatment patters (PMID 38759336).

This novel paper should be discussed PMID 38693018. 

Author Response

Thank you for your review.

Bias in the OS calculation is addressed in the discussion. Unfortunately, the reason of death was not noted, if the death was not from UTUC-cause. The patients dying from bladder cancer, are noted in death from another cause than UTUC. The line added:

  • The OS found in this study is the OS of patients with a diagnosis of LG UTUC treated with laser ablation initially. The OS is relatively low and could be due to the high mean age at diagnosis. Furthermore, this study finds a higher recurrence rate than other similar studies. This could be due to that more patients are selected for local treatment over RNU in this study than in other studies.

Indeed, a subgroup analysis excluding the patients with previous bladder cancer could be interesting to perform. Unfortunately, we only have the number of patients having previous bladder cancer and not which patients that had bladder cancer. The bias is addressed in the discussion instead along the marital status bias, which we unfortunately did not obtain. The line added:

  • Retrospectively a series of data was not obtained from the patients, that could have been interesting to include in the analysis. An important factor that was missed to collect, was which of the patients that had previous bladder cancer, as bladder cancer is known as a risk factor of getting UTUC. This could underestimate the OS, as 30 patients from this study had a history of bladder cancer. A subgroup analysis should have been performed to exclude the 30 patients having a history of bladder cancer. Another factor that we missed to obtain was the marital status of the patients, as this could have had an impact to the effect of the treatment pattern.

Regarding RNU in table 3, these RNU are salvage RNU. RNU performed initially or in the first 3 months from diagnosis are excluded from the study, as shown in figure 1. The text in table 3 is changed to salvage RNU to be more precise.

Regarding comparing RNU with local therapy, indeed this analysis is needed to compare the two treatment methods but requires a whole new study as data from patients undergoing initial RNU were excluded.

Reviewer 2 Report

Comments and Suggestions for Authors

This article provides a well-researched and thorough examination of long-term oncological outcomes in patients with low-grade upper tract urothelial carcinoma (UTUC) treated via endoscopic laser ablation. The authors effectively highlight the significance of this minimally invasive technique, offering compelling evidence of its safety and efficacy, particularly for patients who may not be ideal candidates for radical nephroureterectomy (RNU).

The study addresses a clinically relevant question, exploring laser ablation as a kidney-sparing alternative to RNU for low-grade UTUC patients. In a growing context of organ-preserving treatments, this article delivers timely and valuable insights that could impact treatment protocols. The robustness of the study is emphasized by its large sample size of 118 patients followed over a 10-year period, with up to five years of follow-up, adding credibility to the findings related to recurrence, progression, and survival rates. The use of established statistical tools such as Kaplan-Meier and Nilsson-Aalen plots enhances the reliability of the data analysis. These methods offer clear visual representations of key trends in recurrence, progression, and overall survival (OS), making the results highly relevant to clinical practice. A strength of the study is its balanced discussion on recurrence and safety. While acknowledging the relatively high risk of local recurrence (median recurrence time of five months), the authors point out that the risk of disease progression remains low, and overall survival rates are promising. This careful interpretation reflects the authors’ transparency and comprehensive evaluation of both benefits and limitations.

The findings suggest that laser ablation can be a safe and effective alternative to more invasive treatments, particularly for patients needing to preserve kidney function. This study is especially valuable for urologists and oncologists seeking evidence-based alternatives to RNU for low-grade UTUC patients. In conclusion, the article provides strong support for the efficacy of endoscopic laser ablation, while also stressing the necessity of close post-operative monitoring. Its well-structured methodology and balanced discussion offer clinicians a solid foundation for considering less invasive treatments with favorable long-term outcomes

Comments on the Quality of English Language

.

Author Response

Thank you for your comment. 

Reviewer 3 Report

Comments and Suggestions for Authors

The article could benefit from a more comprehensive introduction that explains the background of low-grade upper tract urothelial carcinoma (UTUC) and laser ablation. For instance, providing statistics on the incidence of UTUC and highlighting why endoscopic laser ablation is preferred for low-grade cases would set a stronger context for the study. This would provide readers, especially those unfamiliar with UTUC, a better understanding of the relevance and importance of the study.

Consistency in terminology should be ensured. For example, the term "local recurrence" should be consistently used throughout, without switching between "recurrence" and "local recurrence" unless distinguishing between different types of recurrences.

The description of the methods is very brief and lacks details. It mentions that data were collected retrospectively and that survival analysis was performed, but it doesn’t explain how data were collected, the type of laser used, or the exact parameters for the procedures.

While the abstract discusses recurrence, it does not specify how patients were monitored post-treatment. Given the high risk of recurrence, it would be useful to know the follow-up frequency and imaging methods used.

The study found significant recurrence rates but modest advancement risks, consistent with earlier laser ablation research on low-grade UTUC. Multiple studies have revealed that local tumor recurrences are common after laser ablation, although survival rates are good, especially when patients are well monitored with frequent follow-ups.

This study had a slightly greater recurrence rate than others. Patient selection or follow-up periods may explain it. The low advancement rate and good renal unit preservation rate match earlier research, where laser ablation is common for low-grade, non-invasive UTUC.

**Conclusion**

The article adds some perspective on low-grade UTUC laser ablation patients' long-term outcomes. The high recurrence rate is alarming, but the low advancement rate and excellent survival outcomes show that laser ablation is a safe and effective treatment for select patients. Maintaining these excellent outcomes requires continuous monitoring and early management upon recurrence.

The study supports laser ablation as a conservative treatment for low-grade UTUC, especially in high-risk patients.

Comments on the Quality of English Language

Minor punctuation issues can be improved for readability. For example, adding commas where appropriate and ensuring consistent use of parentheses for statistical values (e.g., IQR) would enhance flow.

Author Response

Thank you for your comments.

The introduction is now expanded with a more detailed description of UTUC frequency and laser treatment.

The frase local recurrence is now consistent.

For description of methods please see 1) Patient recruitment, where it says that patient information was collected from electronic patient charts. 2) The follow up is described in the diagnosis and treatment plan section. The type of laser is now added in the diagnosis and treatment plan section

Reviewer 4 Report

Comments and Suggestions for Authors In the manuscript, the authors summarized and discussed a single-center series study on endoscopic laser treatment for patients with low-grade upper tract urothelial cancer (LG UTUC). The purpose of the study is to investigate the oncological outcomes of these patients, including recurrence rates, the risk of disease progression, and survival rates. The manuscript is well-organized and clearly stated. I would suggest accepting it after the following minor concerns are addressed: 1.The discussion section should provide a more detailed comparison of the current study's results with those of other studies, especially those that have used different treatment methods. Comments on the Quality of English Language

There are no corrections needed.

Author Response

Thank you for comment. Please see discussion section, where we added several new sections.

Round 2

Reviewer 1 Report

Comments and Suggestions for Authors

The authors addressed properly my comments.